# Consequences of Nephrotic Proteinuria and Nephrotic Syndrome after Kidney Transplant

**DOI:** 10.3390/biomedicines12040767

**Published:** 2024-03-30

**Authors:** María José Ortega, Miguel Martínez-Belotto, Cristina García-Majado, Lara Belmar, Covadonga López del Moral, Jose María Gómez-Ortega, Rosalía Valero, Juan Carlos Ruiz, Emilio Rodrigo

**Affiliations:** 1Immunopathology Group, Nephrology Department, Marqués de Valdecilla University Hospital-IDIVAL, University of Cantabria, 39012 Santander, Spaincovadonga.lopezdelmoral@scsalud.es (C.L.d.M.); rosalia.valero@scsalud.es (R.V.); juancarlos.ruiz@scsalud.es (J.C.R.); 2Pathological Anatomy Department, Marqués de Valdecilla University Hospital-IDIVAL, University of Cantabria, 39012 Santander, Spain

**Keywords:** nephrotic syndrome, nephrotic proteinuria, kidney transplantation

## Abstract

Proteinuria is the main predictor of kidney graft loss. However, there is little information regarding the consequences of nephrotic proteinuria (NP) and nephrotic syndrome (NS) after a kidney transplant. We aimed to describe the clinical and histopathological characteristics of kidney recipients with nephrotic-range proteinuria and compare the graft surveillance between those who developed NS and those who did not. A total of 204 patients (18.6% of kidney transplants in the study period) developed NP, and 68.1% of them had NS. Of the 110 patients who underwent a graft biopsy, 47.3% exhibited ABMR, 21.8% the recurrence of glomerulonephritis, 9.1% IFTA, and 7.3% de novo glomerulonephritis. After a median follow-up of 97.5 months, 64.1% experienced graft loss. The graft survival after the onset of NP declined from 75.8% at 12 months to 38% at 5 years, without significant differences between those with and those without NS. Patients who developed NS fewer than 3 months after the onset of NP exhibited a significantly higher risk of death-censored graft loss (HR: 1.711, 95% CI: 1.147–2.553) than those without NS or those with late NS. In conclusion, NP and NS are frequent conditions after a kidney transplant, and they imply extremely poor graft outcomes. The time from the onset of NP to the development of NS is related to graft survival.

## 1. Introduction

It is well known that proteinuria is the main predictor of kidney graft loss [1,2,3,4] and that it increases the risks of both global and cardiovascular death [2,4,5,6]. Not only the presence of proteinuria but also the quantity is relevant in the surveillance of the graft [4,7], as it correlates to creatinine levels and the corrected measured GFR [8]. The prevalence of proteinuria in transplant recipients depends on the cut-off, varying from 13–36% for an arbitrary established limit of 500 mg/24 h to 50% of the prevalence of microalbuminuria > 150 mg/24 h one year after the transplant [6,9]. High-grade proteinuria (exceeding 1 g/24 h) appears in approximately 4.4–20% of patients 12 months after the transplant [2,6,10] and is mostly due to the loss of albumin, suggesting the presence of glomerular disease in 80% of the patients [1,8,9]. The incidence of high-grade proteinuria (>2 g/24 h) after a kidney transplant in adults has not been well determined and varies from 11 to 22% [11,12,13]. It appears to be more common in children, along with nephrotic-range proteinuria [6,14]. 

Several risk factors may influence the appearance of proteinuria after a kidney transplantation. Most researchers agree that the donor age, delayed graft function, the absence of antihypertensive drugs such as angiotensin-converting enzyme inhibitors or angiotensin receptor blockers (ACEis/ARBs), and an increased recipient BMI can lead to a higher risk of proteinuria [5,7,10,14]. Other risk factors, such as virtual panel-reactive antibodies (vPRA), a donor cause of death, and the existence of donor-specific antibodies (DSA), have been less studied, but have also shown a correlation with proteinuria [2]. 

There are many disorders that can lead to the onset of proteinuria, with some of them being treatable: both chronic and acute rejection, hypertension, transplant glomerulopathy, de novo or recurrent glomerular disease, a polyomavirus infection, and the use of some drugs, for example, imTOR [2,3]. Most of them require the performance of a biopsy to be diagnosed, since there is a low correlation between the amount or type of proteinuria and its cause. The identification of the histological condition may help to establish a prognosis and, in some cases, guide treatment [3,6]. 

There is little information regarding the appearance of nephrotic syndrome in transplant recipients. Few articles can be found [15,16], and they are either quite old [11,12,13,17,18], are single cases [19], or refer exclusively to children [20]. In the experience of our transplant group, nephrotic-range proteinuria after a kidney transplant, defined as >3000 mg/24 h, appears more often as an isolated finding and exceptionally in complete nephrotic syndrome. According to the literature, nephrotic syndrome after a kidney transplant may appear in some cases of FSGS recurrence, membranous nephropathy, and other glomerulopathies [11,19]. In their article from 1980, Cheigh et al. mentioned a 30% prevalence of post-transplant nephrotic syndrome but only referred to nephrotic-range proteinuria as a criterion, without mentioning whether the patients also presented with hypoalbuminemia, peripheral edema, or dyslipidemia [18]. In 1967, Harlan et al. described the clinical and histological profile of 24 patients who developed nephrotic syndrome after a kidney transplant and found edema to be less common than in non-transplant patients [13]. However, this issue has not been fully reviewed recently. 

We hypothesize that full nephrotic syndrome is an exception after a kidney transplant and that, when present, it is a predictor of worse graft outcomes. We studied the clinical profile of kidney recipients who developed nephrotic-range proteinuria to define its clinical and histological characteristics prior to and after the transplantation and evaluated whether the time from the development of high-grade proteinuria to the apparition of hypoalbuminemia was correlated with graft surveillance. We intended to compare these findings between patients with nephrotic syndrome versus isolated nephrotic proteinuria.

## 2. Materials and Methods

An observational, retrospective, single-center study was conducted from January 2000 to September 2023 by following the guidelines of the Declaration of Helsinki, after the study was approved by the regional ethics committee of our institution (reference number: 2023.399PI; 24 November 2023). Among the 1098 kidney transplants performed at Marqués de Valdecilla University Hospital during that period, the data from 204 recipients were analyzed. The inclusion criterion for the analysis was the development of nephrotic proteinuria 6 months after a kidney transplant, defined as proteinuria >3000 mg measured in 24 h collections or >3000 mg/g for the urine protein/creatinine ratio in spot morning urine specimens. Patients were excluded if they had a single determination over this range or two determinations separated by periods without significant proteinuria. Nephrotic syndrome was defined as the development of hypoalbuminemia <3.5 g/dL after the onset of nephrotic proteinuria and before the loss of the kidney graft. 

The analyzed data were retrospectively extracted from the prospectively maintained database of renal transplant patients at our center. For each patient, we collected information regarding the pre-transplant recipient characteristics, the transplant and donor parameters, and the analytical parameters at the onset of NP and at the time of the development of hypoalbuminemia, as described in section Results During the follow-up, the following were noted: the performance of a kidney biopsy; the occurrence of a graft loss, described as the need to re-enter a dialysis program or receive a new transplant; a graft nephrectomy; and death.

Data from the histological reports of recipients who underwent a kidney biopsy within a ±1 year period from the development of NP were reviewed for this study. Some biopsies performed outside this temporary range, both before and after the ±1 year period from the development of NP, were also included. Five patients underwent a biopsy more than one year before proteinuria reached the nephrotic range, but they already showed histological changes that explained the high-grade proteinuria. In eight patients, a graft biopsy was carried out more than one year after the first appearance of NP, mostly due to advanced-stage graft dysfunction suggesting irreversible damage. Since these biopsies aimed to establish the cause of proteinuria, they were also included. The renal transplant biopsies were subjected to light microscopy (LM) with hematoxylin and eosin, periodic acid–Schiff (PAS), Masson’s trichrome stains, and immunofluorescent (IF) studies for IgG, IgM, IgA, C3, C4, kappa, and lambda light chains. The samples were stained using immunohistochemistry for C4d. The 2019 Banff classification for renal transplant biopsies was used for diagnostic categorization. 

The continuous variables were expressed as the mean ± standard deviation if normally distributed, or as the median and interquartile range if non-normally distributed. The categorical variables were described as relative frequencies. Comparisons were achieved using a chi-squared test for categorical variables and Student’s *t*-test or the Mann–Whitney U test for continuous variables, as appropriate. The ability of serum albumin levels to discriminate the risk of developing nephrotic syndrome was analyzed by constructing receiver operating characteristic (ROC) curves, and the optimal cut-off level was established through the Youden index. Univariate and multivariate logistic regression analyses were used to determine the impact of different parameters on graft survival. The graft cumulated survival was compared using Kaplan–Meier curves and a log-rank test. The results were considered statistically significant at *p* < 0.05. The data were analyzed using SPSS statistics software (IBM SPSS Statistics for Windows, version 20, IBM Corp., Armonk, NY, USA).

## 3. Results

### 3.1. Clinical Characteristics

From 1/01/2000 to 1/09/2023, 1098 patients received a kidney transplant at our center, and 204 (18.6%) developed proteinuria >3 g/day (NP) at a median post-transplant time of 55.3 months (20.45–101.69). Among them, 139 (68.1%) also developed nephrotic syndrome (NS) with hypoalbuminemia; therefore, the prevalence of NS after a kidney transplant was 12.6% in our series. The mean patient characteristics and a comparison between those who developed nephrotic syndrome and those who did not are shown in Table 1.The main demographic characteristics did not differ in patients with isolated NP and those who developed NS. The NS group demonstrated a greater prevalence of NS before transplantation, despite the similarity in the cause of end-stage renal disease (ESRD) between both groups. The donor features and transplant-related immunological parameters showed no significant differences between the groups. At the onset of nephrotic proteinuria following the transplantation, individuals with nephrotic syndrome exhibited lower serum albumin levels, although they were still above the lower limit of normal. Low serum albumin levels were able to discriminate the patients who were at risk of developing NS (AUC-ROC: 0.726, 95% CI: 0.654–0.798, *p* < 0.001). The optimal albumin cut-off level was 4.05 g/dL, with a 63% sensitivity and a 73% specificity.

### 3.2. Histology

A kidney graft biopsy for the purpose of establishing the cause of NP was available in 110 (53.9%) patients and was performed at a median time of 44.50 months (13.18–93.82) from the transplant and 1.42 months (−1.26–4.19) from NP development. The histological diagnoses are outlined in Table 2. 

Antibody-mediated rejection (ABMR) was the most common diagnosis (47.3%), followed by glomerulonephritis recurrence (21.8%), especially IgA nephropathy. There were no significant differences in these proportions or the Banff parameters between patients with NP or NS.

### 3.3. Outcomes

The median follow-up time from the transplant was 97.5 months (59.0–147.5). After NP development, 33 (16.3%) patients died, and 131 (64.2%) patients experienced death-censored graft loss (DCGL). The cumulated death-censored graft survival (DCGS) after the development of NP was 75.8% at 12 months, 50.7% at 36 months, and 38.0% at 60 months, as illustrated in Figure 1A. There was no significant difference in the DCGS between patients with or without NS (log-rank, *p* = 0.898), as demonstrated in Figure 1B. In comparison with patients who did not develop NP, the graft survival from the time of the transplant was significantly lower (log-rank, *p* < 0.001) in the study group (Appendix A) and was similar to the patients with NP within 6 months of the transplant (Appendix A).

An antiproteinuric treatment with ACEi/ARB showed no impact on the graft survival, neither in a univariate analysis (HR: 0.689, 95% CI: 0.464–1.024) nor in a multivariate analysis (HR: 0.783, 95% CI: 0.525–1.166) corrected with the GFR. 

Among the 139 patients who developed NS, the median time from the onset of NP to the appearance of NS was 7.28 months (1.28–22.01). The time from NP to NS was related to the DCGL in the univariate (HR 0.997, 95% CI 0.994–1.000, *p* = 0.033) and multivariate (HR 0.996, 95% CI 0.993–0.999, *p* = 0.009) analyses, independently of the GFR (HR 0.961, 95% CI 0.948–0.974, *p* < 0.001).

When analyzed as a dichotomous variable, the 54 patients who developed NS before the third month after NP had a significantly higher DCGL risk in the univariate (HR 1.943, 95% CI 1.308–2.887, *p* = 0.001) and multivariate (HR 1.711, 95% CI 1.147–2.553, *p* = 0.008) analyses compared to the 150 patients with late NS or without NS (100/204), independently of the GFR (HR 0.963, 95% CI 0.951–0.976, *p* < 0.001). The Kaplan–Meier survival analysis showed that early NS was related to a higher DCGL risk (36-month GS: 56.9% vs. 32.6%, log-rank, *p* = 0.001), as shown in Figure 2. 

Similar results were found when exclusively analyzing the patients who developed NS (early NS HR 1.985, 95% CI 1.275–3.091, *p* = 0.002; GRF HR 0.975, 95% CI 0.962–0.989, *p* < 0.001).

## 4. Discussion

The prevalence of nephrotic proteinuria (NP) after a kidney transplant in our series was 18.6%, which fell within the range of 11–22% described in previous studies [11,12,15,16]. Nephrotic syndrome (NS) appeared in 12.6% of our transplant recipients, a rate that is similar to that reported by Yakupoglu et al. in their 2004 series [15] but higher than those of previous reports [18]. This discrepancy may be attributed to the variability in the criteria used to define post-transplant NS. Our higher prevalence might be explained by the retrospective design of our study, where we classified NS based on the presence of NP and hypoalbuminemia, while other authors also included hypercholesterolemia and the presence of edema as criteria. In contrast to Yakupoglu et al., we only included patients who developed NP after 6 months from the transplant. These results suggest that both NP and NS are frequent conditions in our patients, which warrants attention due to their potential consequences.

The mean age in our cohort was 53 years, which is slightly higher than the previously reported age. The male proportion (71.6%) and percentage of cadaveric donors (96%) were also higher in our cohort than the levels previously reported [12,15,16,21]. The distribution of end-stage renal disease (ESRD) mirrored that of the general kidney transplant population, with 38.7% attributed to primary glomerulopathy, which is consistent with the findings in the existing literature [12,15,16,21]. The median time to the onset of NP was 55 months, compared to the previously reported range of 23.7 to 54.6 months [15,22].

The most prominent discovery in our study was probably the exceptionally poor graft outcome among patients with NP, as evidenced by a 64.2% death-censored graft loss (DCGL) after a median follow-up of 97.5 months. The death-censored graft survival (DCGS) after the onset of NP declined from 75.8% by 12 months to only 38% by year five, independently of antiproteinuric treatment being initiated. Compared to the recipients who did not develop NP, the long-term graft survival from the time of transplant was significantly lower in the study group. These poor results closely align with those reported by Yakupoglu [15], who observed a 58% DCGL after a median follow-up time of 80 months, with a 75.3% and 37.5% 1 year and 5 year DCGS in 74 patients with complete after-transplant NS. They also mimic Leal’s results [16], who analyzed a cohort of 50 patients with NP and found a 68% DCGL after a median follow-up time of 93 months. As other authors have described [16,21], the amount of proteinuria above the nephrotic range had no effect on the graft outcomes, which differs from the findings for patients with NP and NS in the context of diabetic nephropathy in native kidneys [23]. Contrary to our initial hypothesis, full NS did not result in worse graft survival compared to NP only. 

The systematic measurement of a recipient’s proteinuria and serum albumin during follow-up visits after kidney transplantation allowed us to analyze whether the time from the onset of NP to the development of NS had repercussions on graft survival in patients with NS. We observed that the 54 patients with early NS (hypoalbuminemia that appeared within three months after NP) exhibited a significantly worse graft survival than the 150 patients with late NS or without NS (Cox multivariate analysis corrected by the GFR, HR 1.711, 95% CI 1.147–2.553, *p* = 0.008). Speculatively, this rapid onset of NS could indicate a higher degree of podocytopathy. Further studies should be conducted to validate these results, as they have not been previously reported. Nevertheless, we believe that identifying this subgroup of patients could guide more aggressive therapeutic measures with the aim of preventing these catastrophic renal outcomes. 

The available data in the reviewed literature comparing patients with NP who developed NS versus those who did not are extremely limited. This topic was only addressed by Cheigh et al. in 1974. However, we found it challenging to draw direct comparisons due to the significant lapse between studies, resulting in differences in study populations, transplantation techniques, immunosuppression, and overall kidney outcomes [17].

Within our group’s findings, we identified two significant differences between the patients who developed NS and those who did not. First, the number of patients who had experienced NS before the transplantation was significantly higher in those who developed post-transplant NS. This observation was previously noted by Cheigh [17]. Second, the NS group already showed lower serum albumin levels at the onset of NP, though within the normal range. In our cohort, albumin levels below 4.05 g/dl at the onset of NP predicted the development of NS with a 63% sensitivity and a 73% specificity. Interestingly, the amount of proteinuria did not differ between the groups, which suggests that low albumin levels could be the result of previous NS or a worse nutritional status at baseline. Some authors have previously observed that higher levels of pre-transplant proteinuria are correlated with higher levels of post-transplant proteinuria, which could predispose individuals to nephrotic syndrome [24]. Higher proteinuria at the time of an ESRD diagnosis has been proven to be a risk factor for certain GN recurrences after kidney transplantation [25], which could provide another possible explanation. Unfortunately, we did not record the patients’ pre-transplant proteinuria levels, but we did record the fact that they had been diagnosed with nephrotic syndrome prior to the transplantation. The Banff parameters and the proportion of ABMR diagnoses in histological specimens were also similar in both groups. The high prevalence of NP and the absence of histological differences between the groups may indicate that the pathogenesis leading to NP after a kidney transplant can be attributed not only to direct podocyte injury but also to hemodynamical changes and hyperfiltration, as observed in diabetic nephropathy [26].

In our study, the reports of 110 graft biopsies were available for review, representing 53.9% of the patients who developed NP. This relatively low proportion in comparison with other cohorts [11,12,15,16] can likely be attributed to many patients developing NP at an advanced graft stage, leading to the procedure being deemed impractical based on a risk/benefit assessment. 

Antibody-mediated rejection (ABMR) was the most common diagnosis (47.3%), with 24.5% of cases demonstrating coexistence with interstitial fibrosis and tubular atrophy (IFTA). In total, 33% of our patients exhibited IFTA changes, though this was the primary diagnosis in only 9.1%. It is worth noting that none of the previous reports utilized the latest revision of the Banff Classification from 2019 [27], making the results challenging to compare. Studies conducted before 2005 referred to chronic allograft nephropathy (CAN), an imprecise and now-abandoned term, as the most common diagnosis of NP and NS after a transplant, with the frequency ranging from 39% to 78%. The majority of these patients would now be classified as having chronic ABMR, along with approximately 20% that would be diagnosed with transplant glomerulopathy [28]. We identified a 7.3% occurrence of de novo glomerulopathy, which is lower than that previously reported [11,15,16], and a 21.8% recurrence of glomerulopathy, which aligns with prior descriptions [15,16,18].

In contrast to other researchers [15,21], who reported worse kidney outcomes when NP was attributed to CAN, our study revealed no significant difference in graft survival based on the histological findings. However, it is worth noting that Ramanathan’s results, indicating poorer outcomes in the CAN group, might have been influenced by a significantly lower glomerular filtration rate (GFR) in that group compared to others, potentially acting as a confounding factor [21]. Additionally, Yakupoglu found the CAN group to have a shorter median time until the development of NS [15], which emerged as a predictor of worse graft outcomes in our study, as discussed below. In alignment with our findings, Leal’s group, whose results were published after the term CAN was abandoned, found no association between the histological diagnosis and the graft prognosis [16]. 

As a retrospective analysis, our study is subject to certain limitations. Due to the reliance on clinical notes for data collection, some information was unavailable. The histological reports exhibited significant variability depending on the timing of the kidney biopsy, in accordance with the Banff Classification in effect during each assessment period. While we acknowledge the importance of providing parenchymal liver enzyme values to assess liver-related defects in albumin production, these data were not included in our study due to the low proportion of patients with relevant hepatopathy in our study group, as proven by the lack of coagulopathy in all patients who underwent a graft biopsy. Nevertheless, we recognize the value of considering these factors in future investigations in this area.

Despite these limitations, there are some strengths to be highlighted. First, our cohort represents the most extensive report of both post-transplant nephrotic proteinuria and nephrotic syndrome, including 204 patients with 110 biopsies analyzed. Second, our research group has consistently conducted systematic analyses of proteinuria and serum albumin levels at every post-kidney transplant visit since the 1980s. This meticulous approach enables us to determine the exact timing of hypoalbuminemia development following the onset of nephrotic proteinuria, a task that is challenging to achieve in non-transplant patients, for whom establishing the initial NP onset time is often unfeasible. Third, our transplant center stands as the sole facility in the region. This ensures an exceptionally low likelihood of missed cases in our series, given our exclusive position as the primary center for transplant-related care in the area. Additionally, few studies have recently reviewed data regarding post-transplant NS [16], and none of them have done so by using the latest Banff Classification from 2019 [27].

## 5. Conclusions

In conclusion, in our series, both nephrotic proteinuria and nephrotic syndrome after a kidney transplant were found to be frequent conditions that imply a poor graft outcome. Graft surveillance does not depend on the development of nephrotic syndrome, the initiation of antiproteinuric treatment, or the amount of proteinuria over the nephrotic range. The main cause of nephrotic proteinuria is antibody-mediated rejection, although the histological diagnosis had no impact on the graft outcomes. Rapid-onset nephrotic syndrome, defined as the development of hypoalbuminemia within three months after the onset of nephrotic proteinuria, significantly worsens graft survival.

## Figures and Tables

**Figure 1 biomedicines-12-00767-f001:**
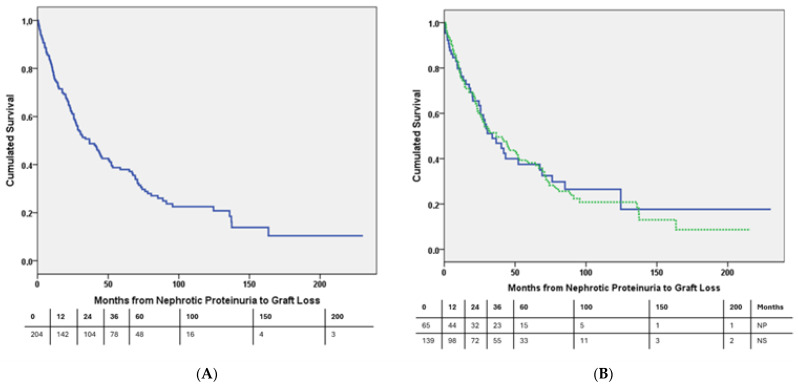
(**A**) Cumulative death-censored graft survival curve of kidney transplant patients with nephrotic-range proteinuria. (**B**) Death-censored graft survival curve. Continuous line represents patients who did not develop nephrotic syndrome, and dashed line represents those patients who developed nephrotic syndrome. Log-rank, *p* = 0.898.

**Figure 2 biomedicines-12-00767-f002:**
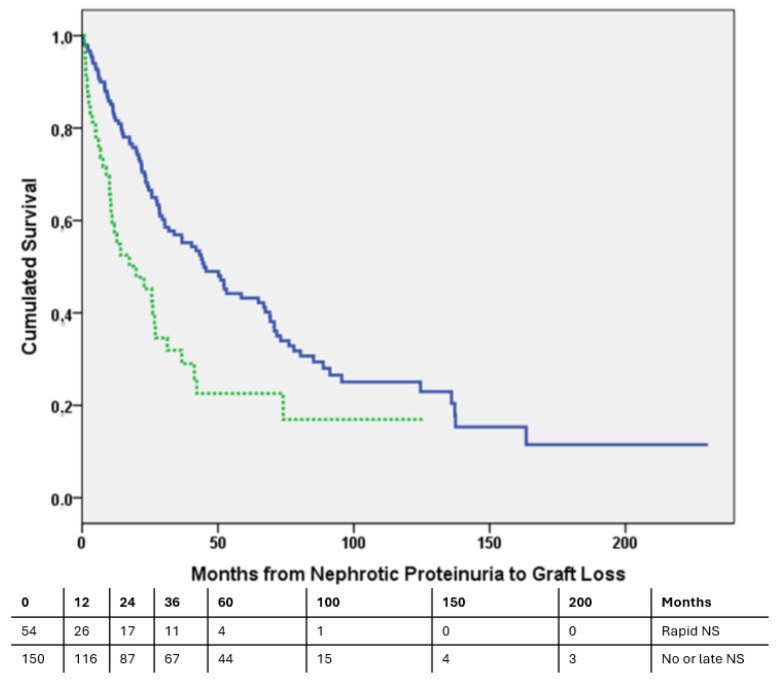
Death-censored graft survival curve. Dashed line represents patients who developed early nephrotic syndrome, and continuous line represents those patients who developed late nephrotic syndrome or did not develop nephrotic syndrome. Log-rank, *p* = 0.001.

**Table 1 biomedicines-12-00767-t001:** Demographic, immunological, and analytical characteristics of patients with nephrotic proteinuria (NP) and a comparison between patients with isolated NP and nephrotic syndrome (NS).

	Total (*n* = 204)	NP (*n* = 65)	NS (*n* = 139)	*p*
Demographic characteristics				
Recipient age (years)	53.1 (39.9–60.8)	49,9 (37.6–58.1)	54 (41.8–61.0)	0.070
Recipient sex (male)	146 (71.6%)	51 (78.5%)	95 (68.3%)	0.136
Pre-transplant diabetes mellitus	55 (27%)	13 (20%)	42 (30.2%)	0.126
Pre-transplant hypertension	184 (90.2%)	56 (86.2%)	128 (92.1%)	0.184
Body mass index (kg/m^2^)	26.4 ± 4.8	26.1 ± 4.1	26.5 ± 5	0.708
End-stage renal disease cause				
Primary GN	79 (38.7%)	28 (43.1%)	51 (36.7%)	
IgA nephropathy	20 (9.8%)	7 (10.8%)	13 (9.4%)	
FSGS	12 (5.9%)	4 (6.2%)	8 (5.8%)	
MPGN	10 (4.9%)	5 (7.7%)	5 (3.6%)	
Membranous nephropathy	7 (3.4%)	2 (3.1%)	5 (3.6%)	
Other	8 (2.5%)	3 (4.6%)	5 (3.6%)	
GN suspected	22 (10.8%)	7 (10.8%)	15 (10.8%)	
Diabetic nephropathy	34 (16.7%)	6 (9.2%)	28 (20.1%)	
Interstitial disease	29 (14.2%)	11 (16.9%)	18 (12.9%)	
Vascular nephropathy	22 (10.7%)	5 (7.7%)	17 (12.2%)	
Cystic disease (ADPKD, nephronophthisis)	15 (7.3%)	6 (9.3%)	9 (6.4%)	
Systemic	9 (4.5%)	2 (3%)	7 (5.1%)	
Other	15 (7.3%)	7 (10.8%)	8 (5.8%)	
Pre-transplant proteinuria disease	147 (73.5%)	45 (71.4%)	102 (74.5%)	0.653
Pre-transplant NS	17 (8.3%)	1 (1.6%)	16 (11.5%)	0.019
ACEi/ARB	161 (78.9%)	47 (72.3%)	114 (82.0%)	0.113
Virtual PRA	12.6 (0–0)	0 (0–3)	0 (0–0)	0.384
Post-transplant NS	139 (68.1%)			
Donor characteristics				
CIT (hours)	20 (16–23)	20 (17–23)	20 (16–23)	0.924
HLA–ABDR mismatches	4 (3–5)	4 (3–5)	4 (3–5)	0.871
Donor age (years)	54.5 (39.7–63.7)	55.5 (44.8–63.3)	55.0 (39.0–64.0)	0.809
Donor sex (male)	116 (56.9%)	31 (47.7%)	85 (61.2%)	0.071
ECD	78 (38.2%)	25 (38.5%)	53 (38.1%)	0.964
Living donation	8 (3.9%)	2 (3.1%)	6 (4.3%)	0.503
DCD	12 (5.9%)	3 (4.6%)	9 (6.5%)	0.652
Immunosuppression and transplant parameters				
Induction	50 (24.5%)	13 (20%)	37 (26.6%)	0.306
Thymoglobulin	40 (19.6%)	13 (20%)	27 (19.7%)	0.923
DGF	64 (31.4%)	18 (27.7%)	46 (33.1%)	0.439
Tacrolimus ^1^	171 (83.8%)	51 (78.5%)	120 (86.3%)	0.155
MMF ^1^	169 (82.8%)	56 (86.2%)	113 (81.3%)	0.391
Steroids ^1^	100 (49%)	32 (49.2%)	68 (48.9%)	0.967
mTOR inhibitor ^1^	52 (25.5%)	17 (26.2%)	35 (25.2%)	0.882
Analytical parameters at the time of the development of NP				
Creatinine (mg/dL)	2.0 (1.6–2.8)	2.02 (1.58–3.1)	1.98 (1.6–2.7)	0.297
GFR (mL/min/0.73 m^2^)	33 (22–44)	33 (20–46)	32 (23–44)	0.597
Serum albumin (g/dL)	4.0 (3.7–4.2)	4.1 (3.9–4.3)	3.8 (3.5–4.1)	<0.001
Serum cholesterol (mg/dL)	184 (153–224)	181 (150–228)	186 (155–221)	0.979
DSA ^1^	32 (15.9%)	10 (15.9%)	22 (15.9%)	0.990
Proteinuria (mg/g or mg/24 h)	3900 (3396–4794)	3726 (3344–4482)	4000 (3388–5225)	0.068
Hematuria	123 (60.6%)	38 (59.4%)	85 (61.2%)	0.263
Hematuria (RBC/HPF)	5 (0–15)	3 (3–10)	5 (0–20)	0.606

^1^ Abbreviations. NP: nephrotic proteinuria. NS: nephrotic syndrome. GN: glomerulonephritis. FSGS: focal segmental glomerulosclerosis. MPGN: membranoproliferative glomerulonephritis. ADPKD: autosomal dominant polycystic kidney disease. PRA: panel-reactive antibody. ACEI: angiotensin-converting enzyme inhibitor. ARB: angiotensin receptor blocker. CIT: cold ischemia time. ECD: expanded criteria donor. DCD: donation after circulatory death. DGF: delayed graft function. MMF: mycophenolate mofetil. GFR: glomerular filtration rate. DSA: donor-specific alloantibody. RBC/HPF: Red blood cell/ high power field. At the time of the development of NP^1^.

**Table 2 biomedicines-12-00767-t002:** Findings in 110 graft biopsy specimens of patients with nephrotic proteinuria after renal transplantation.

Histological Diagnosis	*n*	%
TCMR	5	4.5
ABMR	52	47.3
Pure ABMR	23	20.9
ABMR + IFTA	27	24.5
ABMR + TCMR	2	1.8
ABMR + GN recurrence or de novo GN	6	5.5
GN recurrence	23	21.8
IgA nephropathy	10	9.1
FSGS	3	2.7
Membranous nephropathy	3	2.7
aHUS	3	2.7
Diabetic nephropathy	1	0.9
Membranoproliferative GN	1	0.9
Lupus nephritis	1	0.9
Amyloidosis	1	0.9
IFTA only	10	9.1
De novo GN	8	7.3
FSGS	4	3.6
Membranous nephropathy	3	2.7
Amyloidosis	1	0.9
Other	6	5.5
Total	110	100

Abbreviations. TCMR: T-cell-mediated rejection. ABMR: antibody-mediated rejection. IFTA: interstitial fibrosis and tubular atrophy. GN: glomerulonephritis. FSGS: focal segmental glomerulosclerosis. aHUS: atypical hemolytic uremic syndrome.

## Data Availability

The data presented in this study are available from the corresponding author upon request.

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
