# Peer review of "Consequences of Nephrotic Proteinuria and Nephrotic Syndrome after Kidney Transplant"

_biomedicines, 2024, doi:10.3390/biomedicines12040767_

Round 1
Reviewer 1 Report
Comments and Suggestions for Authors
Renal graft survival represents an important issue, therefore studies focused on this topic could improve our current knowledge. Indeed the presence of significant proteinuria is the main predictor of renal graft loss and subsequently may be linked to an increased risk of cardiovascular mortality. Your study was thoroughly performed, the methodology and results were clearly described, and the conclusions were in agreement with your findings. I congratulate you for your extensive research that highlighted meaningful features of this group of patients, such as the importance of predicting in time the onset of proteinuria (especially in subjects with pretransplant NS, as you have already emphasised) in order to prevent kidney graft loss. Only one minor suggestion - please explain virtual PRA, as you did with the rest of the abbreviations (virtual panel-reactive antibodies).
Reviewer 2 Report
Comments and Suggestions for Authors
The subject of the study is interesting. I have the following comments.
1. Many language errors have to be corrected.
2. In the manuscript, it is repeatedly noted that the biopsies were performed 6 months after transplantation. On the other hand, it is also noted that NS was observed in a range of 1.28 to 22.01 months after transplantation. Why is there such a lag time between the NP and the biopsy? Besides being unusual since biopsy is required for clinical decisions, does it not affect the reliability of the results?
Comments on the Quality of English LanguageMany language errors have to be corrected.
Reviewer 3 Report
Comments and Suggestions for Authors
In the present single-center retrospective clinical study, the authors evaluated how the onset of nephrotic syndrome and serum albumin levels predict graft outcome after kidney transplantation. The paper could be potentially interesting for nephrologists. However, some concerns arise based on the presented data or its interpretation.
First, data of 204 kidney transplanted patients with proteinuria were evaluated out of 1098 total transplanted patients. The major limitation of the study is that, as stated in methods section, the authors excluded all patients who did not develop proteinuria within 6 months aftr Tx, those who could serve as controls. Thus, evaluating only nephrotic patients without such a control group greatly weakens the study.
It would be essential to know, how many percent of the transplanted patients did not get NP or NS within 6 months? Graft survival curves of those non-nephrotic patients is really needed to better evaluate the NP/NS survival curves.
The second problem is that, as also mentioned as study limitation, there is no data about the anti-proteinuric possible medications these patients received. Is it possible that the observed graft survival differences in part are due to differences in RAS blockade, for instance?
What could be the explanation that pre-transplant nephrotic syndrome increased the risk of post-transplant nephrotic syndrome?
In general, the conclusion that lower serum albumin is associated with worse outcome is an obvious pathophysiology, as it reflects more severe proteinuria with higher degree of structural podocyte damage and as a consequence of protein leaking, the latter cause increased tubular protain load that worsens tubular injury.
On the other hand, parenchymal liver enzyme values should be provided as well, to exclude possible liver-related defect in albumin production, and additional hematology parameters (HGB, hematocrit).
Comments on the Quality of English Language
The manuscript is overall well written, easy to follow despite some Enlgish grammar errors and typos that should be corrected (eg. "receptor" should be recipient, NP stands for nephrotic proteinuria, except in discussion where also stands for nephropathy (line 295) - use it for only one term; also line 54-55 "and its cause and"; "cipient sex" in Table 1).
Round 2
Reviewer 2 Report
Comments and Suggestions for Authors
The authors addressed all issues.
Reviewer 3 Report
Comments and Suggestions for Authors
I appreciate the authors' reply and their revision, the paper has been improved.
Just as a minor comment, even in tha absence of electron microscopy one can try to check podocyte damage by performing nephrin or desmin immunostaining on paraffin slides.